# Understanding the Life, Pain, and Barriers to Physical Activity in Korean Patients with Panic Disorder: Photovoice Inquiry

**DOI:** 10.3390/ijerph18158140

**Published:** 2021-07-31

**Authors:** Kyung-O Kim, Jae-Hyeong Ryu, Hae-Ryong Chung, Marcia K. Butler, Deborah Gritzmacher

**Affiliations:** 1Department of Gerokinesiology, Kyungil University, Kyungsan 38428, Korea; 2Chungbuk Boeun Naebuk Public Health Center, Department of Health Administration, Boeun 28943, Korea; jakezzzz526@gmail.com; 3Health and Fitness Management, College of Health, Clayton State University, Morrow, GA 30260, USA; hchung@clayton.edu; 4Health Care Management, College of Health, Clayton State University, Morrow, GA 30260, USA; marciabutler@clayton.edu (M.K.B.); deborahgritzmacher@clayton.edu (D.G.)

**Keywords:** photovoice, patients with panic disorder, life and pain, physical activity

## Abstract

Purpose: This study attempted to understand the life, pain, and barriers to physical activity in the daily life of patients with panic disorder (PD). It aimed to provide specific evidence to promote physical activity for patients, suggesting that suitable physical activity strategies for patients with PD would be of value. Methods: Seven patients were invited to participate in a photovoice study. Photovoice is one example of participatory action research. Results: This study inductively identified two main themes: the life and pain of patients with PD and perceived barriers to participate in physical activity by patients with panic disorder. Conclusion: Based on a specific understanding of the population, this study attempted to provide evidence why patients with panic disorder need appropriate physical-activity-promoting strategies.

## 1. Introduction

Panic disorder (PD) is a type of anxiety disorder accompanied by the physical symptoms of sympathetic nervousness and anxiety about dying, beginning with unexpected and repetitive panic attacks [1]. Panic disorder patients are known to be a very heterogeneous group medically, to have complex pathological causes, and to show various neurophysiological clinical manifestations [2,3]. About 43% of people with PD visit the emergency department and 15% of those use an ambulance to get there. Only 22% of all patients are treated by neuropsychologists, which results in a very low quality of life [4,5]. As a result of analyzing a data set of health insurance patients from Korea National Health Insurance, it was noted that the number of PD patients has increased sharply by an annual average of about 15.8%. The prevalence in 2010 was 51,000 and by 2015 was 106,000 [6]. However, it is estimated that there are far more patients who have not been diagnosed due to social prejudice and rejection of treatment [7].

Treatments for PD in Korea are largely divided into two modalities: medication treatment and non-medication treatment [8]. Treatment utilizing medication has been proven to be effective for some symptoms. Around 95% of the patients are prescribed selective serotonin reuptake inhibitors and other medications including serotonin norepinephrine reuptake inhibitors, noradrenergics, specific serotonergic antidepressants, and tricyclic antidepressants [1,7,9,10]. Cognitive behavioral therapy is one of the non-medication treatments that has been proven to be effective [11,12]. A typical cognitive–behavioral treatment is cognitive restructuring, which focuses on correcting the distorted interpretation of body responses caused by anxiety. Another preferred non-medical treatment is breath retraining, because the most common symptom of PD patients is similar to hyperventilation. In other words, this treatment is focused on breath control [13]. There are other methods such as relaxation, stimulus-sensitive exposure, and actual situation exposure methods. Unfortunately, effects cannot be validated because the accumulation of scientific evidence is insufficient and the cost is also a great burden to patients. Currently, it is known that the prognosis is best when medication treatment and non-medication treatment are combined [14].

Recent clinical studies in North America have looked at the effectiveness of physical activity as an alternative treatment for PD [15,16]. Clinicians reason that physical activity not only lowers anxiety symptoms, but also reduces the main negative physical and cognitive reactions of PD [15]. In addition, it has been reported that physical activity can be an important treatment for PD in terms of its relatively low cost compared to long-term medication treatment and cognitive–behavioral treatment. Furthermore, patients avoid medication side effects and have significant relief of pathological symptoms [17]. Aerobic physical activity is known to be very effective in treating PD. A recent intervention study reported that about 89% of all patients with PD who participated in the study through a physical activity program for about two weeks showed symptom relief [18]. In Korea, most patients still prefer to be treated in hospitals using their National Health Insurance because it costs less. The affordable medical treatments in Korea are different from those in North America, which have a significant impact on patients’ treatment options.

A review of the literature (Google Scholar), using the keyword ‘PD and physical activity’ revealed no results. Of the 50 papers searched with the keyword ‘PD and exercise’, only three related papers were found [19,20,21]. There has not been any research on the possibility of treating symptoms of PD through physical activity. What is known is that the condition of PD as a chronic disease in Korea is very serious, that Korean PD patients cope with their symptoms with health education, and that PD patients who do not receive attention from researchers are socially disadvantaged. Therefore, this suggests that having PD patients participate in physical activity is very important not only to relieve personal pathological symptoms, but also to reduce social costs.

Although PD patients are very sensitive to physical symptoms due to their psychological vulnerability when experiencing negative emotions, it is assumed that they are more likely to show avoidance of participating in physical activity based on the negative impact of their perception regarding physical reactions [22]. The kind of physical activity they participate in and what social and ecological problems they face is not known at all because there is little research on their physical activity. In particular, there is no research on the possibility of an alternative treatment of physical activity for PD. Perhaps Korean kinesiologists need to focus more on this population. The research on physical activity for PD patients should be carried out in the field of kinesiology and, based on the accumulation of this empirical research, we believe that physical activities suitable for PD patients should be developed, disseminated, then researched and published.

In this study, the domestic and foreign academic problems related to ‘PD and physical activity’ are summarized in two points: (1) There are few domestic studies on the physical activity of PD patients, and (2) there is no precise understanding of why PD patients do not actively participate in physical activity even though they know it is effective (how do they know?) for PD.

This study comprehensively analyzes the perceived pain of patients with PD, the meaning of physical activity, the possibility of participating in physical activity, and the problems of participation by tracking aspects of their daily life such as specific emotions, anxiety, and physical symptoms, with photovoice techniques. This study was focused more on investigating subjective emotions and barriers to physical activity and providing detailed evidence to promote physical activity.

## 2. Method

### 2.1. Study Design

This study is participatory action research in the sense that research participants participate in the study through photovoice techniques. This study is also a case study in the philosophical aspect of qualitative research [23] because it explores a very unusual case of physical activity in patients with panic disorder [24,25].

### 2.2. Participant Selection

This study employed purposive sampling to recruit 8 participants. Participants in this study were patients with PD living in Daegu, a large city Korea. From December 2019 to early February 2020, the selection of research participants continued to be discussed with a group of patients with PD (N online community) and a specialist A in the S neuropsychology department of K Hospital who had agreed to cooperate on 15 January 2018. The hospital was positive about recruiting research participants in an indirect way such as posters, since direct recruitment of research participants was not possible as personal medical information is protected in Korea. Eventually, 3 individuals were recruited through the hospital poster, and 12 individuals were recruited through one of popular internet communities, N. However, the criteria (age) of this study prevented 3 applicants from participating in the study, and 4 voluntarily dropped out after explanation of the study. Participant H dropped out of the study due to a simple change of mind.

The study participants were limited to those who were diagnosed with PD in the 30s to 50s age group (the population with the highest prevalence), who still experienced somatization symptoms of PD (irrelevant with or without medications), and who did not have restrictions on physical activity due to other chronic diseases other than PD. For reference, the appropriate number of participants in a photovoice study is known to be around 15 individuals [26]. The general characteristics of the study participants are shown in Table 1 below.

### 2.3. Research Team and Reflexivity

The researchers consisted of three qualitative research experts (one male and two female), one exercise physiologist (male), and a medical doctor (male) who all want to understand the various problems experienced by panic disorder patients. Based on their own specialized experiences, the researchers have rich knowledge and understanding of data collection and analysis for patients with panic disorder. Furthermore, to avoid their specializations from becoming a bias during the study, the researchers conducted discussions and interactions in various ways during the research period. Specifically, Dr. Kim organized and conducted focus group interviews, and MD. Ryu, a medical doctor, accompanied all interviews to minimize the risk posed by interviewing patients. Dr. Kim received training related to qualitative research at the University of Illinois. In particular, Dr. Kim completed his education on the diversity of qualitative research through Dr. Denzin’s class.

### 2.4. Measurement

Using the present photovoice technique, participants can communicate their unique experience through photographs, providing a highly realistic and authentic perspective that is not possible to be understood with traditional qualitative research. In North America, photovoice research has recently expanded to the field of kinesiology [26]. The present study is based on a protocol developed by Baker and Wang and modified by Kim [27], which is divided into three main stages: orientation, photovoice implementation, and focus group discussion.

During orientation, Dr. Kim and MD. Ryu attempted to create rapport with participants and introduced them to all researchers and their achievements. They also had a chance to see pictures of American researchers who were not able to attend the process directly. Ryu, a medical doctor, asked about their symptoms and medications. Through these interactions, researchers and participants were able to get much closer. In a photovoice implementation session, participants were trained on the use of disposable cameras, the decision of what photographs to take, the photograph types they should take, and ethical issues related to taking photographs. Every participant received two disposable cameras, with an approximate capacity of 27 photographs. The red camera was for taking photographs related to pain and the green camera was for taking those related to physical activity. After orientation, participants resumed their daily routines for approximately 2 weeks and took photographs.

### 2.5. Data Analysis

The data in this study were analyzed in parallel with the photovoice analysis method and content analysis. Specifically, participants in the study worked on finding some key codes through entitling their photographs, and researchers conducted content analysis to discover and classify the statements and themes of interview data produced through focus groups.

Through this process, a total of 102 photographs were obtained. The 102 photos were classified as 43 photos related to pain and 59 photos related to physical activity. Research participants also participated in this classification work through pilot interviews and telephone conversations. The seven participants chose a maximum of four photos that were considered important by them. Depending on the researcher’s judgment, duplicate photos were excluded or important photos were added through consultation with the research team and research participants. Finally, this study selected a total of 15 photographs that both research participants and researchers considered most meaningful. A focus group discussion, focusing on these chosen photographs, was conducted with field memos. To guide the focus group discussion, the SHOWeD technique was employed. SHOWeD stands for the following questions: “what do you See happening here?”, “what is really Happening?”, “how does this relate to Our lives?”, “Why does this problem/condition/asset exist?”, “how could this image Educate the community/policy maker, etc?”, and “what can we Do about it?” [26]. Focus group interview data were collected until saturation was reached and all the interview recordings were transcribed. All photographs and interview data were discarded after completion of the research. All focus group interviews were divided into four sessions by two patients, researchers, and doctors due to the government’s ban on having more than five people in a space during COVID-19 restrictions. Each focus group interview took approximately 60 to 90 min. Further interviews for saturation were conducted by telephone due to government quarantine measures under COVID-19.

Interview data from the participants were translated from Korean to English through professional translation services. Firstly, all the participants’ statements were transcribed based on recorded data. Recorded interview files were kept and will be discarded for research ethics after the study is published. Then, a bilingual expert translated all the interview data. Specifically, depending on the language difference, some words were carefully modified for readability based on the expert’s advice. With the interview data, the entire manuscript was revised by an editing service. With those translation processes, the interview could be interpreted and written in the English language.

Researchers read the data from focus group interview and significant statements. The principal researcher organized significant statements with photographs. Then, the research team derived and refined significant statements and themes with photographs through several rounds of discussion. The data analysis process is shown in Table 2 below.

### 2.6. Authenticity and Trustworthiness

The present study attempted to increase authenticity and trustworthiness via triangulation, peer debriefing, negative case analysis, an independent review panel, thick description, and an audit trail. For triangulation, data such as photographs, statements, and field memos were integrated. There were several discussions to share opinions of internal and external experts for peer debriefings. In particular, negative cases, identifying discrepant data, emerged through very few statements expressing positive aspects of physical activity and panic disorder. The researchers then decided not to define these statements as exceptional situations, but to continue to consider and explain the conclusions or limitations of the study [28]. Further, two independent reviewers screened titles of our manuscript and reviewed the full texts. All researchers attempted their best to give a rich description and external experts in the field of qualitative study were asked to audit the process and results of the study, and no particular problems were found. Finally, this study continuously applied the principle of reflexivity to minimize our own influence on the participatory study [28,29].

### 2.7. Compliance with Ethical Standards

This research grant proposal was approved by the National Research Foundation of Korea (NRF) without Institutional Review Board in the year of 2019. However, this researcher informed all research participants of the Helsinki Declaration, especially in all interviews and, to prevent potential risks, a medical doctor was present. In addition, all study participants participated in the study after reading the research statement and signing the research consent form.

## 3. Results

In this study, 15 photographs, 10 sub-themes, and 2 main themes emerged (Table 3). Please see Table 3 below.

### 3.1. The Life and Pain of Patients with PD

Photovoice reveals the life and pain perceived through the eyes of patients with PD. Micro-understanding of the lives of these patients plays an important role in understanding their physical activity patterns.

#### 3.1.1. Fear of Confined or Large Places

Although the titles of the themes seem incompatible with each other, in fact, PD patients have a fear of special places that they define. In other words, narrow and confined places and wide open places are contradictory in terms of semantics, but can be places that give the same feeling to PD patients. The definition of a place where they feel fearful varies from patient to patient, but the place is commonly considered to be a place that you cannot control. That is, if a patient cannot immediately get out of the place, he or she feels anxious there.

The photo in Figure 1 below is an elevator. Three of the patients participating in this study took the same picture of an elevator. This represents a kind of fear of confined places. They say they feel similar anxiety in the park. They are expressing both claustrophobia and agoraphobia at the same time.


*Especially in the elevator… outside, in the park … I feel anxious regardless of where I am…I am worried that the elevator won’t open … Sometime in the past, I don’t know if the elevator was new or not, but it was too quiet. I thought that it broke down. Panic came instantly …*
*(C)*

As already reviewed in previous studies, PD patients have anxiety about specific places. However, this photovoice study was able to demonstrate what places make PD patients feel anxiety and what causes them to feel anxiety or fear through photos and interviews. In addition, some of the anxiety about this particular place is closely related to the second theme, anticipatory anxiety. Although each patient develops anxiety about a specific place that differs from one patient to another, the places were commonly a place that they cannot control.

#### 3.1.2. Anxiety about Somatization Symptoms and Avoidance

Anticipatory anxiety refers to the anticipated anxiety about panic attacks that PD patients usually experience. If a place is avoided due to anxiety about a specific place, it can also be an anticipatory anxiety. In other words, the patients feel anxiety in advance even if the conditions that make them feel anxious are not there. Most of the participants in this study had symptoms of anxiety. This anticipatory anxiety immediately leads to somatization symptoms, which are very diverse. In addition, each individual showed a different mechanism for this anxiety.

The two participants in this study similarly stated through the Figure 2 that they mainly shopped online because it was difficult to go to the market.


*I’m afraid to go to supermarkets or coffee shops, the crowded places, so I order a lot of things online. It was so hard to go to the supermarkets… I went to a large market once, but it was so hard …*
*(G)*

Figure 3 also shows a kind of anticipatory anxiety, which means that certain things are avoided because of a tough memory at a market or a kind of negative thought that they are going to be tough. The following photographs and quotes are statements about avoiding driving due to anxiety.


*I have a precursor before the panic attack comes… When the precursor comes, it’s okay if I take medicine as soon as possible… It gets a lot harder when I cannot do that … I always prepare for the precursor … for everything… Especially if the precursor comes out while driving… it’s very dangerous … so I try not to drive as much as possible… and not to go so far… my spouse usually drives a lot.*
*(A)*

Most of the participants in this study had symptoms of anxiety. Although not all cases can be introduced in this study for internal generalization, it is possible to recognize the tendency of patients with PD. They tended to avoid certain things in advance, either through their experience or through their own negative thoughts. Avoidance of public transportation and memorization of subway maps are additional statements made by the participants in this study with regard to anticipatory anxiety.

#### 3.1.3. Thoughts of Death and Fear

In general, depression is known as a disease that makes the sick want to die, while panic disorder is often described as a disease that makes the patients want to live. However, paradoxically, the fear of death grows because the patients with panic disorder want to live. It also appears that they are always unconsciously recalling their thoughts of death. Figure 4 describes the thought of death.


*The doctors in the hospital always say everyday … that I’ll never die with this disease… But in my everyday life, this disease is always connected to death… Anxiety starts to come …for example, it’s called a panic attack, when I run into it, I just think I will die. I am always afraid of the fear of death… I feel I am going to die.*
*(A)*

When a patient with PD suffers from a panic attack, they can experience dyspnea and fainting, mainly caused by hyperventilation. This is unlikely to lead to death. However, to the patients with PD, the panic attacks experienced by them are scary enough to think that they could die. Even if doctors affirm that they will never die from the disease, the subjective fears experienced by them directly lead to the fear of death. The thoughts and fears of death directly perceived by the patients with PD are very subjective emotions that others cannot easily understand, which significantly lower the quality of the patients’ life.

#### 3.1.4. Limitations of Living and Longing

PD patients have various limitations in living when they live with anticipatory anxiety of a certain place. From a more microscopic viewpoint, there were a lot of minor restrictions in their life. The restrictions make patients think of a healthy life in the past such as having coffee in café shown in Figure 5.


*Now coffee… it contains caffeine. So now, when I talked with the doctor, he told me to avoid having caffeine as well as stress, in my first visit to him… It’s been more than 5 years since I stopped having coffee. But although there are many coffee shops around, I don’t go there… I envy those who go to the coffee shop, and sometimes I want to have a cup of coffee there.*
*(D)*

Most of the patients who participated in this study are talking about longing for the trivial routines they have lost in their life. Panic disorder had a great influence even on their small moments of their daily life.

#### 3.1.5. Side Effects of Medication

Patients with panic disorder usually take a benzodiazepine class of tranquilizers and serotonin reuptake inhibitors for treatment in case of sudden situations. However, these medications cause the patients to experience various side effects which generally occur in all medications. In particular, these medications are psychotropic and are known to have serious side effects compared to other ones. The participant below describes this in detail through Figure 6.


*I have the side effects that I want to do nothing and I just want to sleep… These are psychotropic…When I joined an online community for panic disorder and worked, I saw many people talk the same just as me. If I take medicine, I become lethargic, have little vitality, and go limp.*
*(G)*

The side effects of these medications are described in detail, even if we only search the portal sites. The side effects of these medications for panic disorder, ranging from drowsiness in mild cases to suicidal thoughts in extreme cases, are not to be considered insignificant. Because there are side effects of medications, patients with panic disorder experience great difficulties with the treatment process. Nevertheless, the patients had no choice but to rely on medications.


*I feel comfortable when I take medicine in the hospital… If I don’t have it, I feel anxious. People with panic disorder probably feel like me…*


Medications for treatment are giving the patients with panic disorder a kind of secondary restriction on their life. The patients who participated in this study mainly talked about light drowsiness and lethargy as side effects of medications, but it is assumed that more side effects will be observed if the study is conducted with a larger population group.

#### 3.1.6. Longing That No One Knows

Panic disorder is an invisible disease. In particular, there is a big difference compared to a surgical disease which is clearly visible. Therefore, it is impossible to tell if patients have panic disorder, unless they specifically explain it. The patients also tend to be reluctant to disclose that they have panic disorder. So, they think that everything related to their disease is something that they have to deal with on their own. It is shown in Figure 7.


*This photo was taken with the meaning that I was alone… there is loneliness in my mind. Because other people don’t know and I’m the only one who knows about the disease … When the arms are broken, they can be seen to others, but this disease cannot be seen, so if I don’t explain … I don’t make the opportunities to meet people as much as possible, so I don’t meet people and I feel a bit lonely and a little isolated.*
*(E)*

One participant in this study stated that the reason for communicating through Social Network Services (SNS) is that SNS is the only space where he can avoid the inconvenience of talking about the disease and the various places he does not want to go. This lack of communication is also painful for patients.

### 3.2. Difficulties of Physical Activity Perceived by Patients with Panic Disorder

The problems or difficulties related to physical activity perceived by patients with panic disorder are associated with their life. If their life is well understood, their low levels of physical activity can be understood. Nevertheless, the reason why it is necessary to closely understand patients’ subjective perceptions of physical activity is that physical activity can also become the object of avoidance in their life. Accordingly, the themes in this study are divided as follows.

#### 3.2.1. Avoidance Due to Panic Attacks or Anticipatory Anxiety Experienced during Physical Activity

Some patients with panic disorder who participated in this study stopped doing physical activity after experiencing panic attacks during the physical activity. This point is closely related to the theme in their life of anticipatory anxiety and avoidance of somatic symptoms. Patients with panic disorder had a strong tendency to negatively accept positive physical changes during physical activity. In other words, there was a tendency to connect the changes in heart rate and difficulties in breathing which may usually occur with their disease as shown in Figure 8.


*I don’t want to think about it, but if I do, I went to the emergency room because I thought I would die from hyperventilation while playing soccer… That’s why I don’t play soccer anymore…*
*(E)*

The following participant in this study also made statement similar to the above case.


*I played badminton… I was playing badminton, but my heart suddenly seemed to be beating fast and I couldn’t breathe well… I stopped playing and got help from others … I have been to the emergency room by calling 119.*
*(B)*

Most patients with panic disorder complain of constant breathing discomfort. This is not because there is actually a medical abnormality, but because they perceive it as such, which develops into a somatic symptom. Although most patients with panic disorder recognize the process of panic attacks, anticipatory anxiety occurs because they are afraid of experiencing somatic symptoms. This makes it more difficult to participate in physical activity, especially for patients who are sensitive to somatic symptoms. In particular, the following participant in this study expressed the discomfort he felt during physical activity through a straw in Figure 9.


*I haven’t exercised, but sometimes I’ve tried … Whenever I exercised … I felt like I was breathing with this straw in my mouth. This means that I am out of breath. Now I cannot overcome this, so it’s very difficult to exercise.*
*(G)*

#### 3.2.2. Happy Memories, but Now I Can’t …

Most of the patients who participated in this study had their own physical activity routines such as walking. However, since the onset of their panic disorder, most of their old routines were broken. For them, the routines related to physical activity in the past could not be done now and they have become a happy memory. The following participant in this study used a tennis racket as in Figure 10 to explain that he enjoyed participating in physical activity in the past, but now he does not.


*I took a picture of a tennis racket … It’s been a while since I had panic disorder … before that, I really liked playing sports. So, the happiest time for me is when I exercise, and I miss those times a little … and … I haven’t used the racket since I had panic disorder although I have it in my car … So, whenever I open the truck and look at the racket, I think it was really good time in the past, but why is it like this now?*
*(F)*

As illustrated above, things that seem meaningless to someone can have a completely significant meaning to a specific person, a patient with panic disorder in this study. This is an important role of photovoice.

The participants in this study consistently recalled memories of participating in physical activities that were enjoyable in the past. At the same time, they felt a kind of loss that they could not do them now. It becomes easier to understand when this point is connected with their life. They experience a variety of restrictions in life, and physical activity is a part of these restrictions. It is natural for the participants to miss the things they had given up in their past life, and they remembered the physical activity which they had given up as a happy moment.

#### 3.2.3. Anticipatory Anxiety about Places of Physical Activity

As shown in the lives of patients with panic disorder, they have a tendency to avoid certain places. Naturally, the anticipatory anxiety caused by direct experience or subjective thoughts ultimately affects their physical activity.

As we have understood through their life, a space to them has a subjective meaning, completely different from what ordinary people think. One participant in this study expressed that even a large park was a space that made him anxious.

The participant is referring to the shower room in Figure 11 that he naturally goes to after physical activity. That is, he stated that the shower room that he has to go to after physical activity is a kind of narrow space causing anxiety. Because this aspect is also very subjective, more diverse results may be obtained when a large number of subjects are studied. However, a diverse range of places related to physical activity are not comfortable for patients with PD.


*This is a photo of a shower room … After exercising, I must take a shower. Because it is hard for me to enter such a narrow place … I won’t do it… when I exercise, because I have to take a shower again, I have to go into a narrow space … that is continuously connected, to me …*
*(B)*

#### 3.2.4. Lack of Desire to Exercise

The following participant in this study tried to express his helplessness and indifference to physical activity through a photo of a gravesite in Figure 12.


*It seems that not only the desire for exercise but also the desire for diverse lifestyle often disappears. I feel like I am drowsy and thoughtless …*
*(D)*

After being diagnosed with panic disorder, patients gradually get tired of treatment with medication and various other treatments. The patients particularly feel helpless with all other pathological symptoms, when the disease is not cured even with the treatment that they have tried intentionally or selectively. The participant in this study also stated that he had been receiving outpatient treatment and psychological treatment for a long time, but he had a hard time with not getting better. He also said that the desire for physical activity itself disappeared as a result.

#### 3.2.5. Nevertheless, Physical Activity Is a Challenge

Doctors in hospitals, where patients with panic disorder regularly visit, encourage the patients to do physical activity. Therefore, the patients try to do physical activity to treat their disease although they are afraid. Not all attempts to do physical activity are satisfactory, but there is hope for treatment in their mind. The participant below regularly checks his heart rate through a smart watch shown in Figure 13. In particular, he shows his will to do the physical activity by monitoring his heart rate during physical activity such as walking. Similarly, another participant in this study talks about the existence of a partner as shown in Figure 14. These two participants have different ways, but they can be understood in the same context that they know their pathological condition and have a safety device for themselves.


*I bought a smart watch … because the heart rate can be checked with it … because I keep looking at the heart rate. Now I check the watch when I feel a little breathless and anxious … if my heart rate is normal, I think I feel relieved. I have a habit of always wearing the watch and checking it whenever I move.*
*(C)*


*When I exercise, there must always be a lot of people … if there is no one, I don’t exercise. I need a partner for any other reason… well… I need people around me … if I’m in an emergency, they can take me to the emergency room or treat me … and yes … so actually I cannot exercise often.*
*(F)*

As shown in Figure 15, the following participant in this study remembers or checks the location of the hospital as usual. In particular, he has a habit of checking the location of the hospital on his first trip. When it comes to physical activity, this task seems even important.


*It means that I check a location of a hospital in order to go there quickly when I have a panic attack. Of course, I know the hospital on the way I always go, and if I go a different route for the first time, I always check the nearby hospital first before I start to go. It is the same with exercise. Even if I cannot participate in sports sometimes, if I go with my friends, I think that I can do it because I like playing sports and I am good at it … I check the location of a hospital that I can get to as soon as possible and then I go to watch the sports.*
*(A)*

## 4. Discussion

### 4.1. To See the World through Their Eyes

The present study aimed to understand a patients’ life, pain, and perceived barriers to physical activity and provide basic data to promote physical activity in PD patients. The seven participants were physically inactive, which can be ascribed to their unique pathological symptoms. This photovoice inquiry focused on two overarching themes: the meaning of life and pain, and perceived barriers to physical activity. First, “Fear of confined or large places” can be regarded as a personal issue that exerts negative effects on the daily lives of many PD patients [30]. For them, the meaning of place is completely different from that of ordinary people and is very subjective [31]. The fact that any place can be a scary place for patients is expressed through the picture of an elevator and the spacious park picture. Therefore, it is very important to understand that feelings about place are what they decide subjectively [32].

The second theme is “Anticipatory anxiety about somatization symptoms” focused on the lives of many PD patients. All the patients in this study reported anticipatory anxiety [33,34]. In the present study, patients could express the extent of their psychological pain via the photovoice method. Patients reported anticipatory anxiety about specific actions in specific places and used the imagery of the supermarket and driving to describe their psychological pain and its negative effect on their lives. We believe that the promotion of physical activities to patients will cause secondary pain if no attempt is made to understand their daily lives [35].

Third, “Thoughts of death and fear” focused on thoughts of death from invisible fear. Although doctors call it an indestructible disease, patients paradoxically express the nature of this disease, where patients feel fearful enough to die [36]. The thoughts and fears of death that PD patients directly perceive are very subjective feelings that others cannot easily understand, and this greatly lowers the quality of life of patients [37]. The pains they experience every day have made them naturally compare their daily lives to a grave.

Fourth, “Limitations of life and longing” showed that patients complain of loneliness because of the invisible characteristics of neuropsychiatric diseases and socially negative views about these diseases [38]. As the cup of coffee they once enjoyed merges with the doctor’s advice associated with caffeine intake and their own fears, their daily lives are being destroyed. Accordingly, they miss their daily routine before the diagnosis of PD.

Fifth, “Drug side effects” means that the patients are experiencing severe side effects compared to other medications. Side effects appear differently depending on the type of medication, but side effects such as drug dependence, insomnia, fatigue, aggression, decreased concentration, decreased memory, drowsiness, and suicidal thoughts mainly appear in the drugs commonly consumed by PD patients [9,39,40]. The side effects differ depending on the type and dose of the drug, and this should be thoroughly discussed with a doctor. Particularly, when PD and depression are comorbid, special attention to drug intake is required. These drug side effects also reduce their quality of life.

Sixth, “A loneliness no one knows” is a typical psychological reaction experienced by patients with PD. PD is very different from surgical disease. PD is perceived only by the person and has no external symptoms, not like if a leg was broken [41]. Due to these characteristics, it is difficult for people around them to recognize patients with PD, and the patients are reluctant to expose their disease due to the negative image of PD in Korea. The six sub-themes that have been shown represent the daily distress of PD patients.

### 4.2. Patients with PD and Barriers to Physical Activity

The second major theme was “Barriers to participate in physical activity perceived by patients with PD”. Patients with PD have been shown to be unable to participate in physical activity regularly due to various psychological and pathological symptoms [42,43].

The first sub-theme was “avoidance due to panic attacks experienced during physical activity and anticipatory anxiety”. Most Korean patients with PD are encouraged to do physical activities at the same time as medication (participant I). However, people with PD tend to associate increased heart rates and difficulty breathing with their illness, which are common symptoms during physical activity [44]. This prevents them from participating in scientifically proven physical activities [35,45], or they participate in limited physical activities with anxiety. One participant likened breathing discomfort to a straw. These reasons restricted participation in physical activity of patients with PD.

Second, “Happy memories, but things that can’t be done now” means that patients’ happy routines have been broken. One patient recalls their happiest moment through a tennis racket and their happy past when they saw a tennis racket in the trunk of their car. Such an ordinary object can have a special meaning for patients with PD. At the same time, patients feel a sense of loss that they cannot do it now [46]. This is the important role of photovoice [26].

Third, “Anticipatory anxiety about the place of physical activity and avoidance”. In the case of panic attacks during physical activity, the patient experiences anticipatory anxiety about the place. For example, patients who experience panic attacks on a soccer field feel nervous when they go to a soccer field. In particular, anticipatory anxiety is known to be an important predictor of quality-of-life changes [47]. Therefore, lowering anticipatory anxiety through a variety of treatments can be an important factor in allowing patients with PD to participate in physical activity [48].

The fourth sub-theme is “Lack of desire for physical activity”. Patients are slowly exhausted as they experience medication and various treatments after they are diagnosed with PD [49]. This phenomenon may be divided into two categories: fatigue from long-term treatment and lethargy from drug side effects. These two factors have shown that patients have less desire for physical activity.

The final sub-theme is “Nevertheless, physical activity must be challenging”. People with PD are aware of the effects of physical activity but are unable to exercise due to diverse reasons. Some participants in this study used smart watches or participated in physical activities with a partner. These behaviors are positive signs of the desire to participate in physical activities and at the same time they experience desire to protect oneself at the top of the list. This serves as important information in constructing strategies to promote physical activity in patients with PD.

Therefore, this study suggests that the composition of physical activity strategies for patients with PD should be prioritized to fully understand the pathological characteristics of patients, because PD patients are very heterogeneous.

### 4.3. Phyical Activity for Patients with Panice Disorder

The prevalence of PD in Korea has been rapidly increasing recently. The number of patients in Korea diagnosed with PD has rapidly increased from 50,000 in 2010, to 100,000 in 2015 and 144,000 in 2017, but patients are not receiving proper treatment due to social prejudice and negative perception of personal psychiatric treatment [1]. Many empirical studies of PD have been produced in Korea and other countries, but the results of these studies have not dramatically led to the promotion of physical activity in patients with PD. Although PD is a neuropsychiatric disease, physical activity is bound to be limited by individual symptoms because various physical symptoms are expressed. Further, Korean doctors recommend physical activity, but no recommendation is made based on individual symptoms.

### 4.4. Symptoms in People with Panic Disorder Vary Widely, Making It Very Difficult for Doctors to Recommend Certain Physical Activities. I. Neuropsychiatrist

Thus, the accumulation of various symptoms recognized by patients with PD is very important academically. In particular, if studies produced in various academic fields do not contribute to changing patients’ daily lives and patterns, these studies remain research studies. However, this study explored the lives, pain, and barriers to physical activity of patients experiencing PD together to produce an empirical study, reducing negative social phenomena. The advantage of this study is that it can be useful in evaluating, improving, or developing physical activity in the PD population. In other words, it is very necessary to present a more appropriate alternative by exposing the invisible problems and pain associated with their physical activity to the world, and it is even more important to prove it scientifically.

Although not all studies have shown consistent results, many studies show that lower physical activity in patients with PD can generally lead to diverse diseases such as obesity, and it is easier for women to become obese. Further, one study found that people who take antipsychotics used in panic disorder typically sleep more than nine hours a day, with a body mass index above 30 kg/m^2^. That is evidence that these patients are susceptible to obesity [50]. Therefore, it is important to produce a delicate physical activity strategy for these patients.

This research determines that expanding social interest in the lives and physical activities of the socially disadvantaged, as well as those with PD, and the production of repeated and continuous follow-up studies are essential. The accumulation of micro-understanding may also be used to establish institutional mechanisms for patients with PD.

## 5. Limitations

The present study is subject to the following limitations. As social qualitative research, the present study neither examines any medical or bio-physical causation using a treatment, nor was conducted by medical personnel. Thus, it was not possible to collect any medical information from the research participants. With regard to the number of participants, only seven patients were invited to this study. Moreover, the photos selected by patients may be subjective, and may not be represent the whole PD population in South Korea. Thus, the results of the present study may be useful for patients with PD in similar circumstances, as this study pursued internal generalization, a characteristic of qualitative research. Nevertheless, it is clear that the results of this study are not applicable to all PD patients. Furthermore, this study reveals that there were very few statements explaining positive relationships between physical activity and panic disorder.

## 6. Conclusions

Although some empirical studies on physical inactivity of patients with PD have been conducted [43,51], there have been insufficient studies on the existing barriers of everyday life and physical inactivity in PD patients.

This photovoice study revealed life, pain, and constraints associated with physical activity as perceived by patients with PD. This photovoice study presented the invisible problems and pains that PD patients face with regard to their life and physical activity. Furthermore, this methodological approach provided important interpretation for understanding populations with PD, improving patient quality of life, and providing policy implications. This study explored practical strategies for enhancing quality of life and physical activities of PD patients, in order to ultimately provide micro-understanding of the patients. There is a need to accumulate understanding of various invisible pains in more patients to improve their health and quality of life and reduce economic and social costs. In the future, more valuable results will be produced if photovoice studies are conducted on larger populations of patients with PD.

## Figures and Tables

**Figure 1 ijerph-18-08140-f001:**
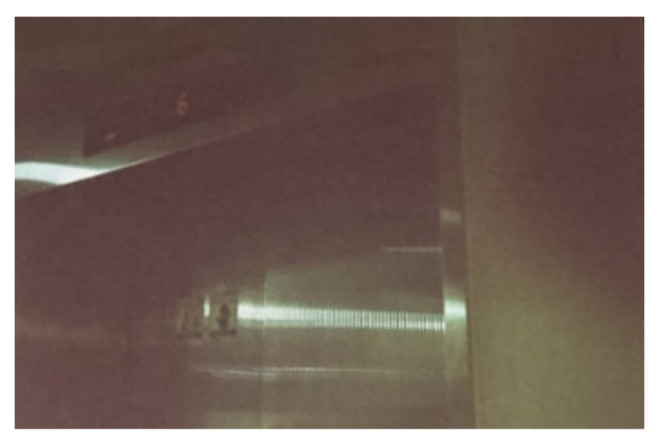
Elevator.

**Figure 2 ijerph-18-08140-f002:**
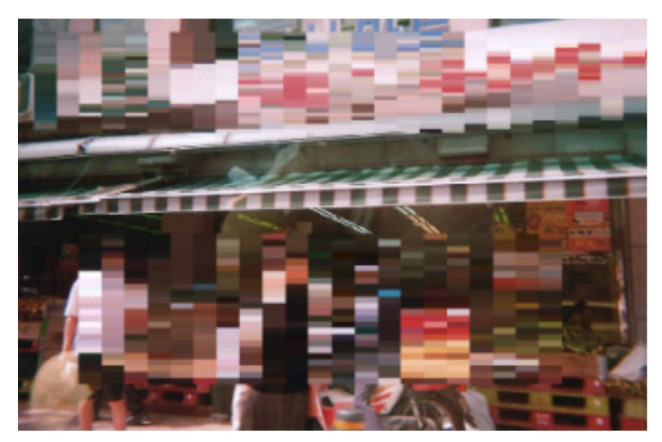
Supermarket.

**Figure 3 ijerph-18-08140-f003:**
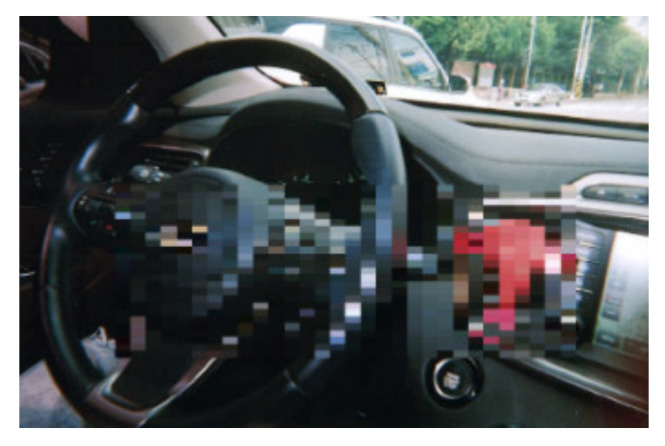
Driving.

**Figure 4 ijerph-18-08140-f004:**
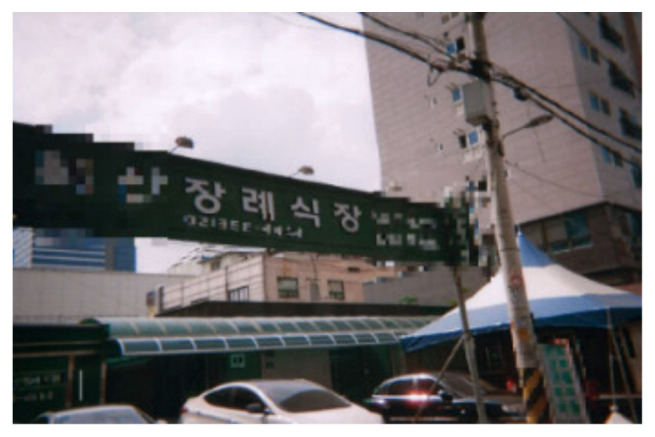
Funeral hall.

**Figure 5 ijerph-18-08140-f005:**
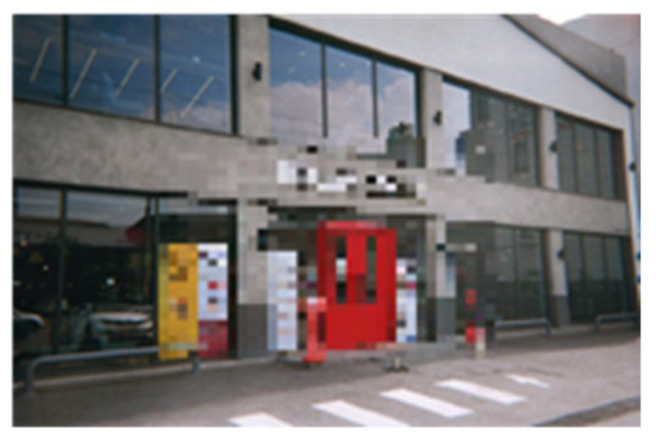
Cafe.

**Figure 6 ijerph-18-08140-f006:**
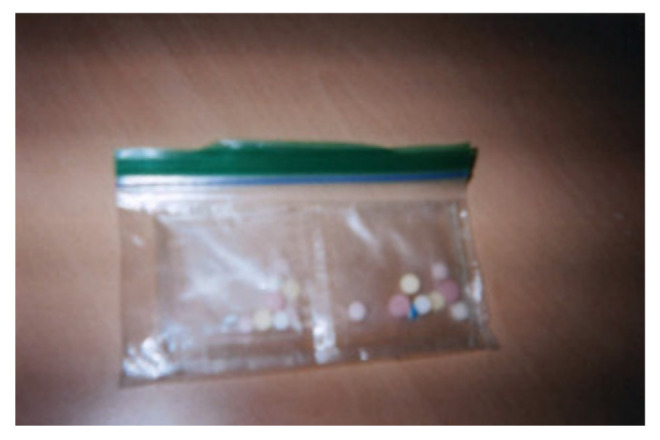
Medications.

**Figure 7 ijerph-18-08140-f007:**
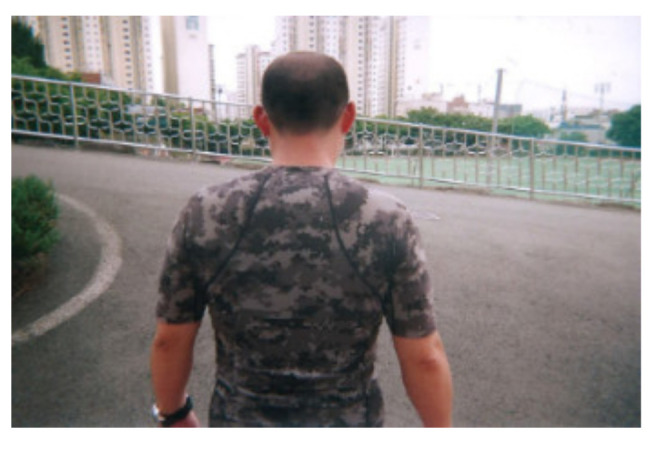
My back.

**Figure 8 ijerph-18-08140-f008:**
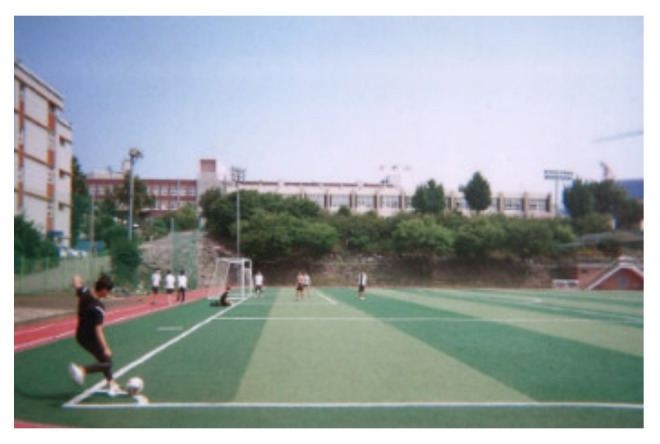
Soccer field.

**Figure 9 ijerph-18-08140-f009:**
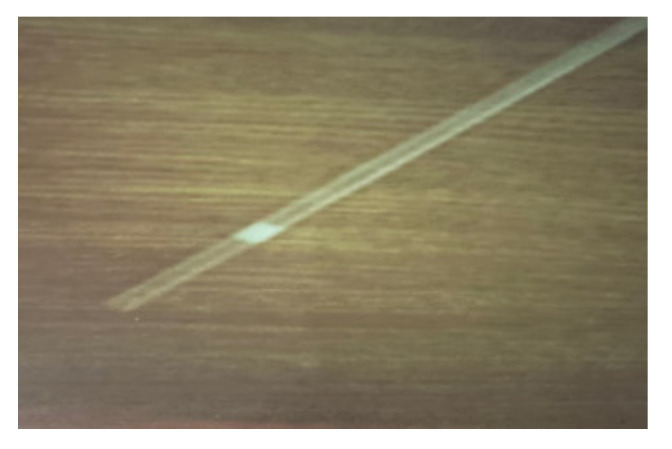
Straw.

**Figure 10 ijerph-18-08140-f010:**
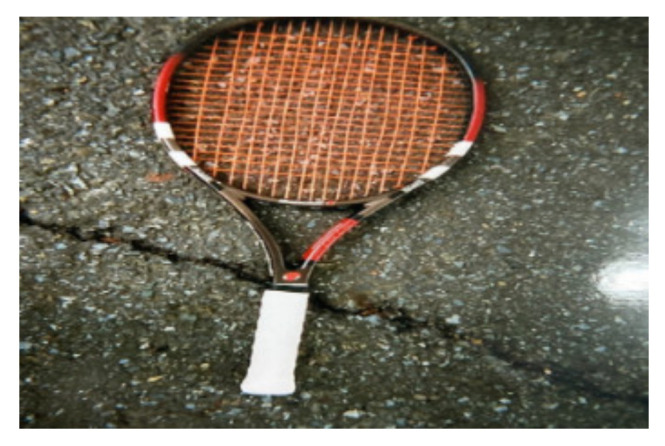
Tennis racket.

**Figure 11 ijerph-18-08140-f011:**
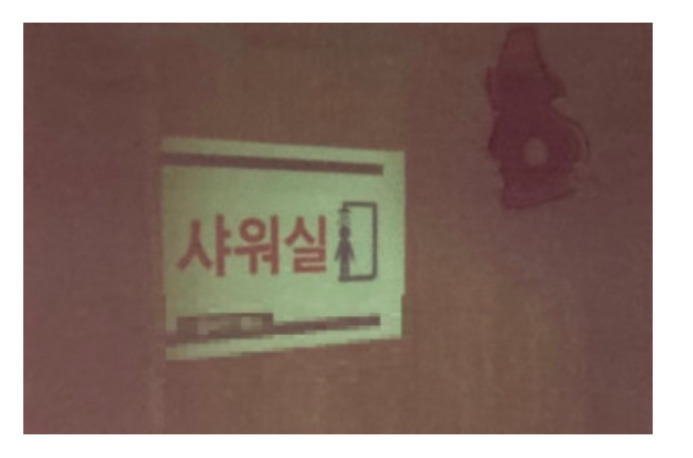
Shower room.

**Figure 12 ijerph-18-08140-f012:**
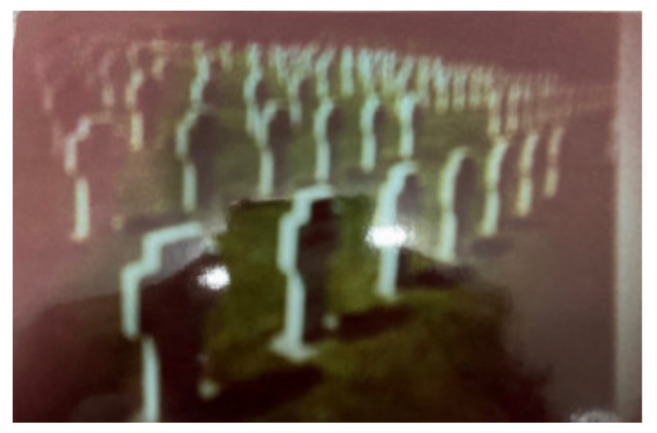
Gravesite.

**Figure 13 ijerph-18-08140-f013:**
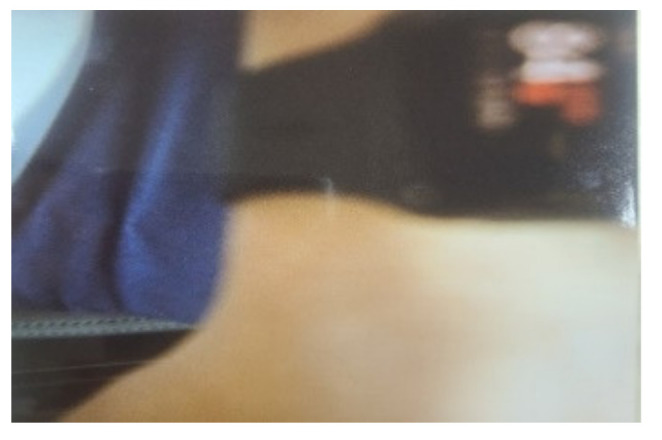
Smart watch.

**Figure 14 ijerph-18-08140-f014:**
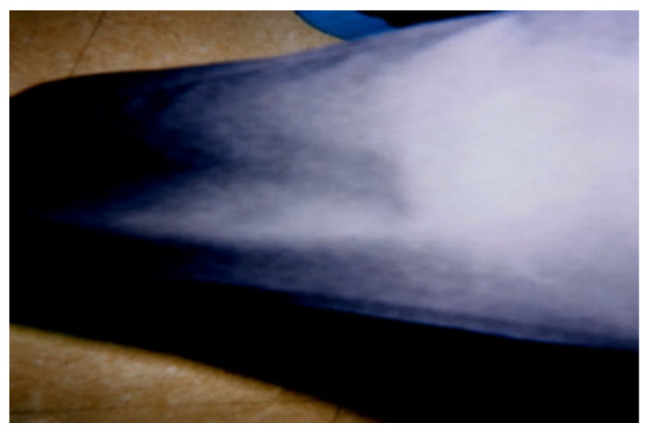
Partner’s leg.

**Figure 15 ijerph-18-08140-f015:**
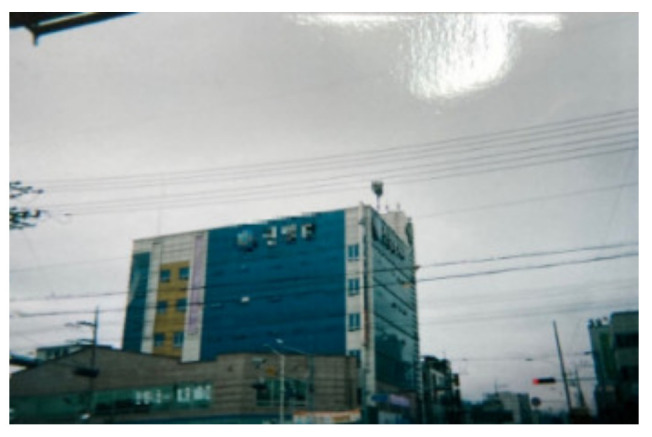
Hospital.

**Table 1 ijerph-18-08140-t001:** General characteristics of participants.

Participant	Gender	Age	Marital Status	Family/ Inmate	Monthly Income (USD)	Education Level	Regular PA Participation	Remarks
A	Male	52	Single	Alone	1500 or over	College graduation	NO	
B	Male	42	Married	Children	1500 or over	College graduation	NO	
C	Male	54	Married	Spouse	1500 or over	High school graduation	NO	
D	Female	38	Single	Relative	500 or over	Grad school graduation	NO	
E	Male	42	Married	Spouse	1500 or over	College graduation	NO	
F	Male	46	Married	Spouse	1500 or over	College graduation	NO	
G	Female	42	Married	Spouse	1500 or over	College graduation	NO	
H	Female	34	Married	Single	500 or over	High school graduation	NO	Drop-out
I	Male	47	Married	-	-	Neuropsychiatrist	-	Medical Advice

**Table 2 ijerph-18-08140-t002:** Data analysis process.

Stage	Details	Operator
1	Finding significant statements with photographs	All researchers
2	Integration, organizing and collecting significant statements with photographs	Kim, Buttler, Gritzmacher
3	Classifying categories (e.g., pain and physical activity) through the classification of important statements	Kim, Buttler, Gritzmacher
4	Finding main themes and sub-themes in separated categories	All researchers
5	a) Repeated discussion and reflection of the research team to determine the final theme b) Holistic analysis based on overall statement, descrption, theme, and interpretation c) Sharing data with research participants for theme confirmation	All researchers & participants
6	Creating report	Kim

**Table 3 ijerph-18-08140-t003:** Presentation of the obtained results.

Main Theme	Sub-Themes	Title of Photos
The life and pain of patients with PD	- Fear of confined or large places	- Elevator
	- Anxiety about somatization symptoms, and avoidance	- Supermarket - Driving
	- Thoughts of death and fear	- Funeral Hall
	- Limitations of living and longing	- Cafe
	- Side effects of medication	- Medications
	Longing that no one knows	- My back
Difficulties of physical activity perceived by patients with panic disorder	- Avoidance due to panic attacks or anticipatory anxiety experienced during physical activity	- Soccer Field - Straw
	- Happy memories, but now I can’t …	- Tennis racket
	- Anticipatory anxiety about places of physical activity	- Shower room - Gravesite
	- Nevertheless, physical activity is a challenge	- Smart watch - Partner’s legs - Hospital

## Data Availability

The data presented in this study are available on request from the corresponding author. The data are not publicly available due to protection of patients’ privacy.

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
