# Peer review of "Understanding the Life, Pain, and Barriers to Physical Activity in Korean Patients with Panic Disorder: Photovoice Inquiry"

_ijerph, 2021, doi:10.3390/ijerph18158140_

Round 1

Reviewer 1 Report

Dear Authors,
I congratulate you on taking up the topic.
The research issue is important and will probably grow in importance.
The more noticeable is the lack of clear (also in graphical form) presentation of the obtained results.
Undoubtedly, making the form of presentation of results more attractive would not only have a cognitive value for the reader but would also provide a valuable premise for a potential scientific discussion with other researchers.
Please consider introducing the presentation of results in the form of graphs.
Regards

Reviewer 2 Report

This manuscript entitled "Understanding the Life, Pain, and Barriers to Physical Activity in Korean Patients with Panic Disorder: Photovoice Inquiry" aimed to scrupulously understand the life, pain, and barriers to physical activity in the daily life of patients with panic disorder (PD) .

The manuscript is very interesting. However, some issues should be addressed by the authors:

Major issue:

  • Very small sample size and does not allow to make solid conclusions. I suggest the authors to improve the sample size.

INTRODUCTION

  • This sections is laking of clarity and flow. Please, improve the rationally.
  • Try to keep shorter and more objective.  Maybe 5 or 6 paragraphs are enough.

METHODS

  • Please, improve data analysis. How did the authors analyzed data?

REFERENCES

  • Introduction and discussion should be updated with articles from the last 5 years.
  • Please, several articles from IJERPH could be cited in the introduction. Please, go through the Journal.

Reviewer 3 Report

The manuscript is very interesting and novel. The introduction is sufficient, the methodology used is adequate and consistent with the purpose of the study, and the results support the discussion. However, I have some comments.

I. Major Comments:
1. Physical activity influences body weight. I suggest briefly discussing this point.
2. Is there an increase in body weight in people with PD? It would be interesting to discuss this point.
3. Considering their lifestyles, do people with PD present a greater risk of developing obesity? Also, has the diet of people with PD been studied? have a healthy diet?

II. Minor Comments:
1. Improve wording of the study objective
2. Correct writing errors
Example: page 2, line 40. "PD (with or ........" replace by "PD (with ......"
3. The name of the city is S?

Round 2

Reviewer 2 Report

All my comments were addressed by the authors. Congrats!